**Data Availability Statement:** All relevant data are within the manuscript and its Supporting Information files.

# Histopathologic patterns and factors associated with cervical lesions at Jimma Medical Center, Jimma, Southwest Ethiopia: A two-year cross-sectional study

**Birhanu Hailu Tirkaso**[1]*, **Tesfaye Hurgesa Bayisa**[2], **Tewodros Wubshet Desta**[3]

**1** Department of Biomedical Sciences, Mizan-Tepi University, Mizan, Southwestern Region, Ethiopia, **2** Department of Pathology, Jimma University, Jimma, Oromia, Ethiopia, **3** Department of Pathology, Dire Dawa University, Dire Dawa, Ethiopia

* birhanuhailu@mtu.edu.et

## Abstract

### Background

The cervix is the lower portion of the uterus, which connects this organ to the vagina through the endocervical canal.

### Objective

This study aimed to determine the histopathologic patterns and factors associated with cervical lesions at Jimma Medical Center from September 12, 2017, to September 12, 2019.

### Methods

A 2-year facility-based cross-sectional study was conducted from May 1 to June 30, 2020.

### Result

In this study, cervical cancer was the most common (71%) cause of cervical lesions. Squamous cell carcinoma was the most frequent cervical cancer diagnosed during the study, accounting for 96.4% of 331 cancerous cases, followed by adenocarcinoma (3.3%). High-grade squamous intraepithelial lesions were the most frequently diagnosed precancerous lesions, accounting for 68.4% of cases. Endocervical polyps were the most commonly diagnosed benign lesions, accounting for 59.3% of cases.

### Conclusion

The maximum age distribution of cervical lesions was in the 41–50-year age range. Squamous cell carcinoma was the most frequent type of cervical cancer. High-grade squamous intraepithelial lesions were the most frequently diagnosed precancerous cervical lesions. The most common benign cervical lesions were endocervical polyps.

**Funding:** The author(s) received no specific funding for this work.

**Competing interests:** The authors have declared that no competing interests exist.

## Recommendation

We recommend educating the community to improve health-seeking behavior and on possible preventive strategies for cervical cancer.

# 1. Introduction

## 1.1. Background

The endocervical canal connects the uterus to the vagina through the cervix, which is the bottom part of the uterus. Non-keratinizing squamous epithelium covers the majority of the exocervix. A layer of columnar mucus-secreting cells makes up the glandular mucosa of the endocervix, and the intersection between the squamous and glandular epithelia is called the squamocolumnar junction [1, 2]. The squamocolumnar junction undergoes eversion with the onset of puberty, causing columnar epithelium to become visible on the exocervix. The exposed columnar cells eventually undergo squamous metaplasia, forming a region called the transformation zone [3]. The transformation zone is the area where the expected transformation to the metaplastic squamous epithelium and the abnormal transformation to CIN usually occur [4].

The range of neoplastic conditions of the cervix arises at or near the cervical transformation zone, ranging from precancerous intraepithelial neoplasia to epithelial malignancies, mixed epithelial and mesenchymal tumors, mesenchymal tumors, melanocytic tumors, and miscellaneous rare malignancies [5].

The epidemiology of precancerous lesions and squamous cell carcinoma (SCC) of the cervix follows the epidemiology of HPV infections. In developed countries, adenocarcinoma now makes up 10–25% of all cervical carcinomas, up from 5–10% thirty years ago. This rise is due to a decrease in squamous cell carcinomas brought on by screening programs and improved glandular lesion detection in cervical cytology samples. It is advised to employ a two-tier system of low- and high-grade intraepithelial lesions rather than the three-tier CIN 1, CIN 2, and CIN 3 terminology since it is more biologically applicable and histologically repeatable. The pairing of CIN 2 and CIN 3 as an HSIL is supported by our enhanced understanding of HPV biology. The typical type of SCC can be either keratinizing or nonkeratinizing [6].

According to a 5-year retrospective study done on 75 cases in Nigeria, cervical cancer has an incidence of 65.2% of all gynecological cancers and 13.4% of all gynecological admissions. The modal age was in the 60–70 years range. SCC of varying differentiation was the most common histological type, accounting for 89.3%, and adenocarcinoma accounted for only 8.0%. The modes of presentation were post-menopausal bleeding (84.0%), vaginal discharge (72.0%), contact bleeding (63.9%), and abdominal pain (56.2%) [7].

## 1.2. Problem statement

The overall burden of cancer is thought to be increasing globally, especially in developing nations. Thirteen million people will pass away from cancer and twenty-one million will receive a diagnosis by the year 2030. Even while mortality rates are just 8% to 15% higher in more industrialized nations, incidence rates for all malignancies combined are more than twice as high in more developed nations compared to less developed ones. This mismatch mostly reflects variations in diagnostic and therapeutic modalities' accessibility and availability. Similar to how tumors are more frequently found in the advanced stages in less developed

nations, which contributes to the discrepancy in fatality rates compared to incidence. The National Cancer Control Plan (NCCP) of Ethiopia has established challenging goals to advance preventative intervention techniques, introduce screening exams for early detection, and provide chemotherapy, surgery, and palliative care in diagnosis and treatment. As seen in industrialized countries, it takes years for these measures to reduce the incidence of cervical cancer significantly [8].

Cervical cancer is one of the most common cancers in women, with approximately 0.5 million cases worldwide in 2008. With an average annual growth of 0.6%, cervical cancer cases grew from an estimated 378,000 in 1980 to 500,000 per year in recent years. Roughly 76% of recent cases occur in low-resource countries, with numbers rising everywhere but in high-income nations. In countries with greater affluence, the median age at death is 55. Effective programs to detect cancer precursors along with infrastructure to clinically manage precursor lesions are largely responsible for the significant differences in incidence and mortality between the more severely affected nations in Africa and Asia and the lower rates in wealthier North American and European countries [6, 9].

Cancer is one of the major non-communicable diseases (NCDs). Together with NCDs, cancer causes over 60% of total global mortality every year. Although communicable diseases remain the leading killers, the incidence and mortality from NCDs are rising rapidly in many developing countries. This has resulted in a double disease burden, which is putting strain on the existing health system [8, 10]. According to GLOBOCAN 2018, the global cancer burden has risen to 18.1 million cases and 9.6 million cancer deaths. One in six women and one in five men worldwide develop cancer during their lifetime, and one in eleven women and one in eight men die from the disease. The incidence of cancer worldwide is expected to rise to 26.4 million, with 17 million deaths by 2030 [11, 12].

A 2-year retrospective study done on 3231 samples at UOG: North-West Ethiopia showed that 540 (16.7%) were cancer cases. Cervical cancer and breast cancer were the two most common cancer types in the hospital, with cervical cancer ranking first [13].

In 2015, an estimated 21,563 and 42,722 incident cancer cases were diagnosed in males and females, respectively, according to a study based on primary data on 8539 patients from Addis Abeba's population-based cancer registry and supplemented by 1,648 cancer case data from six regions. Cervical cancer was the second most common and prevalent cancer, making up 23% of cancer cases, preceded by breast cancer (33%) [14].

Cervical cancer accounts for 22.5% of all cancer cases in women in sub-Saharan Africa, and the majority of women who develop cervical cancer live in rural areas. WHO estimated that cervical cancer will kill more than 443,000 women per year worldwide by 2030, nearly 90% of them in Sub-Saharan Africa [15].

### 1.3. Rationale of the study

In Ethiopia, cervical cancer is the second most common female cancer. Throughout the world, HPV infections are the most common cause of cervical cancer. Out of more than 100 different HPV types, infection with HPV 16 and 18 has been associated with more than 70% of cervical cancers. Cervical cancer is a preventable and curable disease. It can be prevented by vaccination and screening and cured if identified early.

Even though high-risk HPV infection is a significant cause, only a small percentage of HPV-infected women develop invasive cancer, implying the presence of HPV infection co-factors that lead to cervical cancer development. According to studies carried out in different parts of Ethiopia, including the study area, knowledge about cervical cancer is very low. Although the NCCP launched nationwide HPV vaccinations in the 2016–2019 national cancer

control plan, the effects take years to be seen in primary and secondary school students. Furthermore, PAP smears are only performed in a few centers across the country, and other health institutions in the country's outskirts perform a visual inspection with acetic acid (VIA). To my knowledge, no studies have included benign cervical conditions in the study area. The clinicopathologic studies done in the study area were conducted ten years ago.

### 1.4. Significance of the study

This study augments the clinicopathologic studies done in southwestern Ethiopia from a histopathologic point of view, including benign cervical conditions. In addition, this study will serve as a baseline for future studies to be carried out at the national or continental level during the post-HPV vaccination era. The result of this study also serves as input for JMC, Zonal and Regional Health Bureaus, and the Ethiopian Federal Ministry of Health (EFMOH) in evaluating and monitoring the screening and preventive strategies.

## 2. Methods

### 2.1 Study area

The study was conducted at Jimma Medical Center, formerly called Jimma University Specialized Hospital (JUSH). JMC was established in 1938 in Jimma Town, located 352 kilometers southwest of Addis Ababa, the capital of Ethiopia. Jimma is the main town in southwestern Ethiopia with an estimated population of 120,960. It has a latitude of $7°\,41'\,6''$ North, a longitude of $36°\,49'\,53''$ East, and an elevation of 1738 meters (5702 feet). Currently, the hospital is a teaching and referral hospital in the southwestern part of the country, providing service for approximately 15,000 inpatients, 160,000 outpatient attendants, 11,000 emergency cases, and 4,500 deliveries in a year, coming from the catchment population of about 15 million [16–18].

The pathology department is one of the most important departments at JMC, providing services such as histopathology diagnostics, FNAC, fluid cytology, and hematopathology, with an annual average flow of 1,928 histopathology biopsy samples, over 5,000 FNAC, and over 200 BMA.

### 2.2 Study design and period

A 2-year facility-based cross-sectional study was conducted from May 1 to June 30, 2020.

### 2.3 Population

**2.3.1 Target population.** All female patients in the catchment area of Jimma Medical Center.

**2.3.2 Source population.** All female patients who submitted biopsy specimens to the pathology department for histopathology diagnosis from September 12, 2017, to September 12, 2019.

**2.3.3 Study population.** Histopathology reports of all female patients who submitted cervical tissue specimens to the pathology department from September 12, 2017, to September 12, 2019.

**2.3.4 Study unit.** Histopathology reports of selected female patients who submitted cervical tissue specimens to the pathology department from September 12, 2017, to September 12, 2019, and fulfilled the inclusion criteria.

## 2.4 Inclusion and exclusion criteria

**2.4.1 Inclusion criteria.** All female patients who submitted cervical tissue specimens to the pathology department from September 12, 2017, to September 12, 2019.

**2.4.2 Exclusion criteria.**

- Biopsy reports missing two or more independent variables

- Biopsy reports with inconclusive histopathologic diagnosis

- Repeated biopsy

## 2.5 Sample size and sampling technique

Histopathology hard copy reports of 543 patients at JMC's pathology department from September 12, 2017, to September 12, 2019, were retrieved from the pathology department's data archive, and those reports that fulfilled the inclusion criteria were manually selected and then grouped by year after being retrieved from the entire hard copy of 3766 histopathology reports in the department's archives. Relevant information was collected via a checklist from 469 hard copies. See Fig 1.

## 2.6. Study variables

**2.6.1. Dependent variable.**

- Histopathologic pattern of Cervical tissue specimens

**2.6.2. Independent variable.**

- ♠ Age ♠ Type of biopsy

- ♠ Place of Residence ♠ Clinical features

## 2.7 Data collection procedures

Histopathology reports of biopsies submitted from cervical lesions that were routinely processed and from which paraffin sections were taken and stained with hematoxylin and eosin

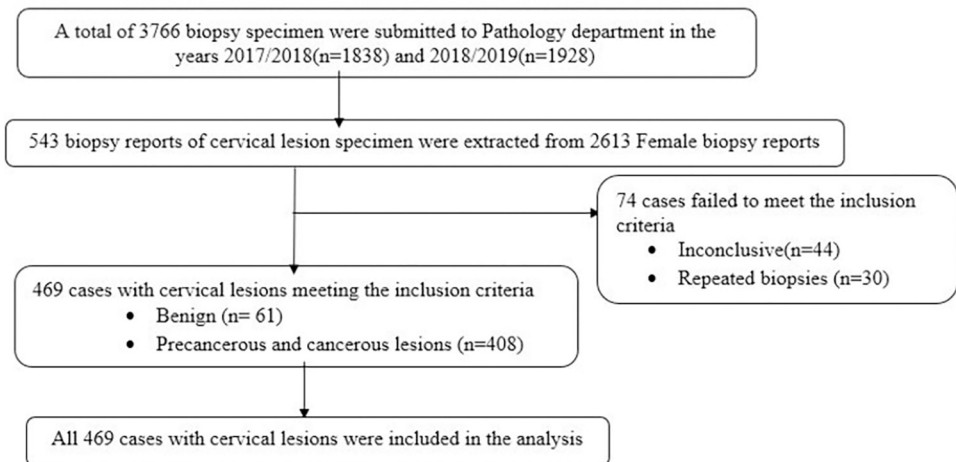

**Fig 1. Diagram showing the sampling procedure of the selected 469 cervical lesion biopsy records from 2017 to 2019.**

(H&E) for microscopic examination and histopathological diagnosis at JMC, pathology department, from September 12, 2017, to September 12, 2019, were retrieved from the pathology department data archive. Eligible 469 reports fulfilling inclusion and exclusion criteria were extracted and recorded into a prepared checklist containing study variables. All necessary preventive methods for coronavirus disease-19 (COVID-19) were used throughout as per the national protocol by using personal protective equipment, physical distancing, and disinfectants.

## 2.8 Data processing and analysis

Data were cleaned, coded, entered into EpiData v3.1, and exported to SPSS version 26 for analysis. Cross tabulation, the chi-square test, and logistic regression with multivariate analysis were performed to look for associations between the study variables. The logistic regression model was utilized to model a linear relationship with the logit of the outcome. The regression coefficients represent the intercept (b0) and slope (b1) of the line. When solving this equation for the probability (P), the probability has a sigmoidal relationship with the independent variable (X) [19].

$$\ln\left(\frac{p}{1-p}\right) = b_0 + b_1 X$$

Variables with P values less than 0.5 were chosen for multivariable logistic regressions. A p-value of 0.05 was used as a cutoff point for identifying predictors for histopathologic patterns. The findings were presented using text, tables, and charts.

## 2.9 Data quality management

A checklist was adopted after reviewing different kinds of literature and books, and the checklist was pretested on 47 cases (10% of the total sample size) of hard-copy biopsy reports done in the year 2016 that were not included in the current study. Then the checklist was revised with some modifications to the variable, and the final revised checklist was used for data collection. Two days of training were given to the data collectors on how to locate, retrieve, categorize, and record the data. The principal investigator subsequently followed and supervised while the data collectors were retrieving and recording the biopsy results from the pathology department data archive using checklists. Following a thorough review of the checklist, data were entered into Epidata on a password-protected computer and exported to SPSS version 26 for analysis.

## 2.10 Ethical consideration

Before data collection, the proposal for this study was submitted to Jimma University's research and ethical committee. Then, ethical clearance for the use of secondary data and a waiver of informed consent was obtained from the Institutional Review Board (IRB) of JMC and submitted to the responsible authorities of JMC. A formal letter of cooperation was also written to obtain permission from the head of the pathology department before using the data. All the information collected from the study was handled confidentially by omitting the personal identification of the pathology reports, and data utilization was carried out by the ethical and scientific standards outlined in national and international guidelines.

## 2.11 Operational definitions [1, 6]

Cervical tissue specimens: specimens of patients with cervical complaints such as punch biopsies and hysterectomy specimens.

Cancerous cervical lesion: malignant lesions of the cervix.

Histopathologic pattern: a specific type of diagnosis made on biopsy specimens.

Noncancerous cervical lesion: benign conditions of the cervix.

Precancerous cervical lesion: precursor lesions of cervical cancer (LSIL and HSIL)

## 2.12 Limitations of the study

Biopsy results are not computerized and compiled with the clinical data of patients; thus, most of the factors associated with cervical cancer cannot be assessed. Confirmatory immunohistochemistry and HPV DNA testing could not be included in this study because of the limited setup. Power analysis for sample size calculation was not performed.

## 3. Results and discussions

### 3.1 Results

In 2017–2018 (n = 1838) and 2018–2019 (n = 1928), a total of 3766 biopsy specimens (2613 female and 1153 male) were submitted to the pathology department. From 2613 female biopsy reports, 543 biopsy reports of cervical lesion specimens were extracted, of which 74 were excluded because they fulfilled the exclusion criteria. Cervical biopsy specimens accounted for 14.4% of the total biopsy specimens and 20.8% of the total biopsy specimens from female patients who submitted biopsy specimens during the study period.

**3.1.1. Sociodemographic profiles.** Of the 469 cervical lesion cases, 244 cases (52%) were from 2017–2018, and 225 (48%) cases were from 2018–2019 (Fig 2)

The age distributions have a minimum value of 22 years and a maximum value of 85 years, with a mean age of 47.06 years and a standard deviation of 11.019 years. The maximum age distribution of cervical lesions was in the 41–50-year age range, accounting for 154 (32.8%) biopsies, followed by 31–40 years with 143 (30.5%) biopsies, and 51–60 years with 102 (21.7%) biopsies.

More than half of precancerous and cancerous cervical lesions (65.46%) occurred after the age of 40 years, while benign lesions accounted for 47.5% after the age of 40 years. This was statistically significant. ($X^2$ = 23.54, DF = 5 $p$ = **0.001**).

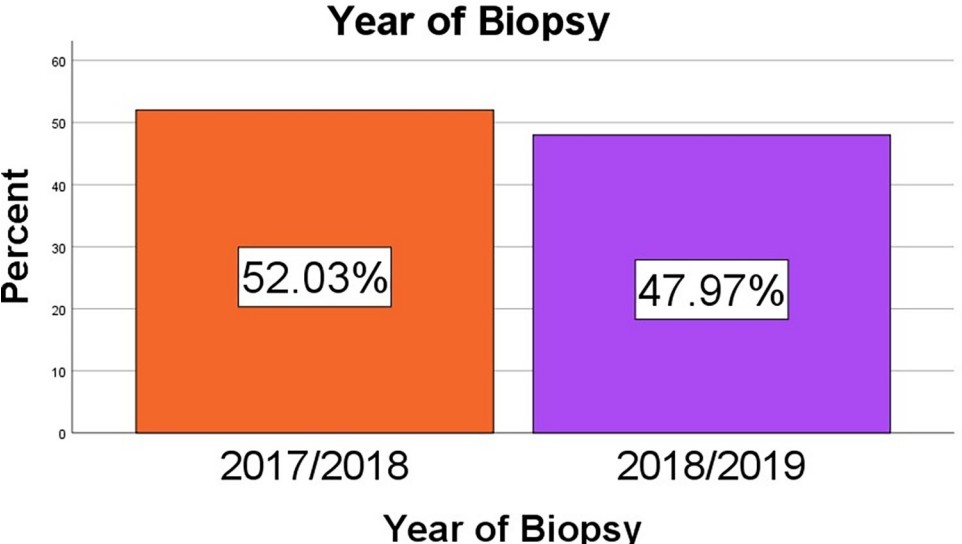

**Fig 2. Bar graph showing the distribution of cervical lesions between 2017 and 2019.**

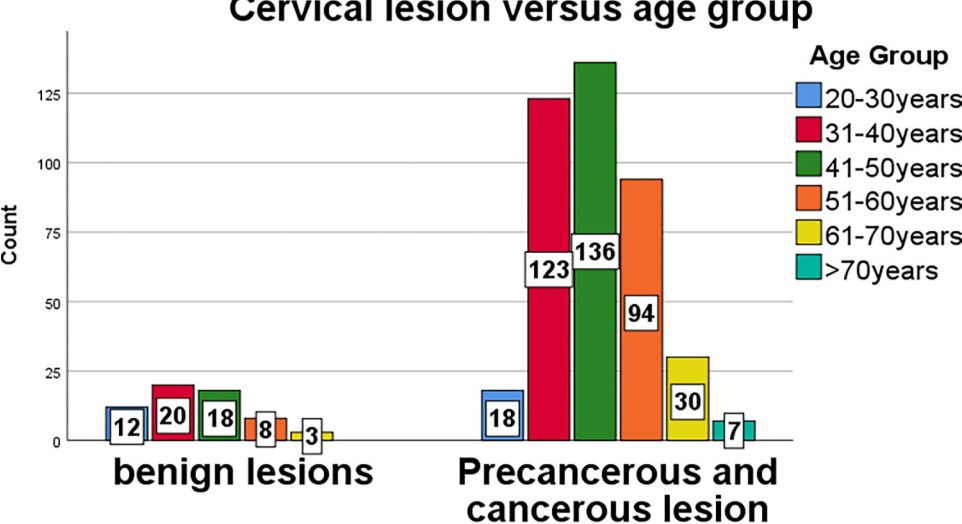

**Fig 3. Bar graph showing cervical lesion status with age groups.**

As the patient's age increases, the risk of precancerous and cancerous cervical lesions increases, and the third and fourth decades of life were found to be associated with an increased risk of having precancerous and cancerous cervical lesions (Fig 3).

Most of the patients (77.8%) were from surrounding areas with variable distances from Jimma town, while 104 (22.2%) were from Jimma town. The majority of patients coming from the periphery were from the Jimma zone, accounting for 58.4% (Fig 4). Most (88.8%) of the cases from the surrounding areas of Jimma were precancerous and cancerous cervical lesions, while 80.8% of the cases in Jimma town were precancerous and cancerous cervical lesions. This showed a strong association between precancerous and cancerous cervical lesions and residency. (X2 = 4.67, DF = 1, p = 0.027).

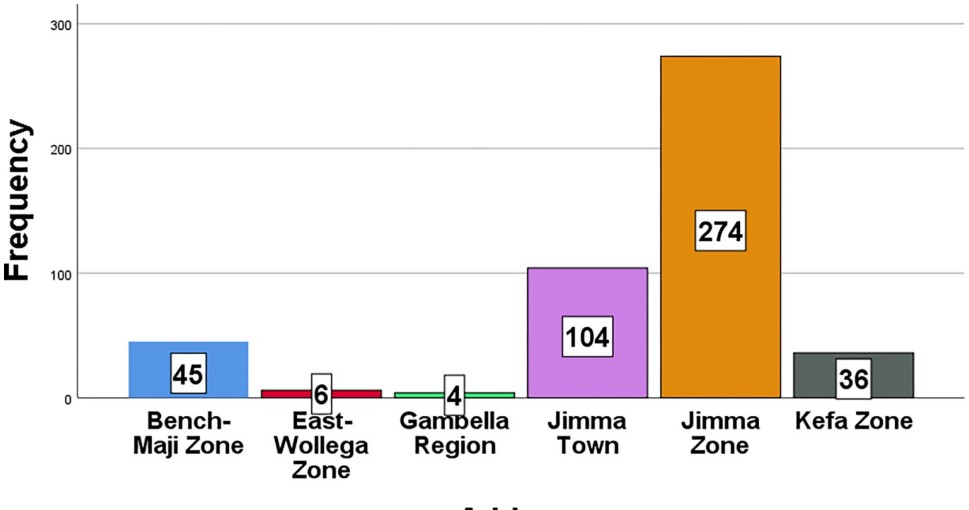

**Fig 4. Bar graph showing the distribution of cervical lesions by address.**

**Table 1. Clinical features/presentation of patients with cervical lesions.**

| Clinical features/presentation | Frequency | Percent |
|---|---|---|
| Vaginal Bleeding | 226 | 48.2 |
| Vaginal Bleeding and Vaginal Discharge | 108 | 23.0 |
| Vaginal Bleeding and Lower abdominal pain | 89 | 19.0 |
| Vaginal Discharge | 44 | 9.4 |
| Cervical CA Screening | 2 | 0.4 |
| Total | 469 | 100.0 |

**3.1.2. Clinical data of patients with cervical lesions.** The most common presenting symptom of patients with cervical lesions was vaginal bleeding alone, accounting for 206 (43.9%) patients, followed by vaginal bleeding with vaginal discharge (108 (23%) patients) and vaginal bleeding with lower abdominal pain (89 (19%) patients) (Table 1). The duration of symptoms in patients with cervical lesions ranged from four days up to sixteen years (Fig 5). The most common type of biopsy performed for patients with cervical lesions was a punch biopsy, accounting for 91.3% followed by abdominal hysterectomy, accounting for 6.4% (Table 2). Most (90%) patients with cervical complaints were clinically diagnosed with cervical cancer (Fig 6)

**3.1.3. Histopathologic patterns of cervical lesions.** Most (87%) of the specimens were precancerous and cancerous cervical lesions, while 13% were benign cervical lesions (Fig 7). The majority (71%) of cases were cancerous cervical lesions, followed by precancerous cervical lesions (16.4%) (Table 3). Squamous cell carcinoma (SCC) was the most frequent cervical cancer diagnosed during the study period, accounting for 96.4% (Table 4). HSIL was the most frequently diagnosed precancerous lesion, accounting for 68.4%. Endocervical polyps were the most commonly diagnosed benign lesions, accounting for 59.3% (Table 3). The most common histologic subtype of SCC was keratinizing SCC, accounting for 77.4% of all cases, while the remaining 22.6% were nonkeratinizing SCC (Fig 8).

Cancerous cervical lesions tend to develop in older age groups, and benign conditions tend to develop in younger age groups. SCC occurred most frequently in the 41–50-year age group (N = 104), followed by the 31–40 year (N = 86) and 51–60-year age groups (N = 85) (Table 5).

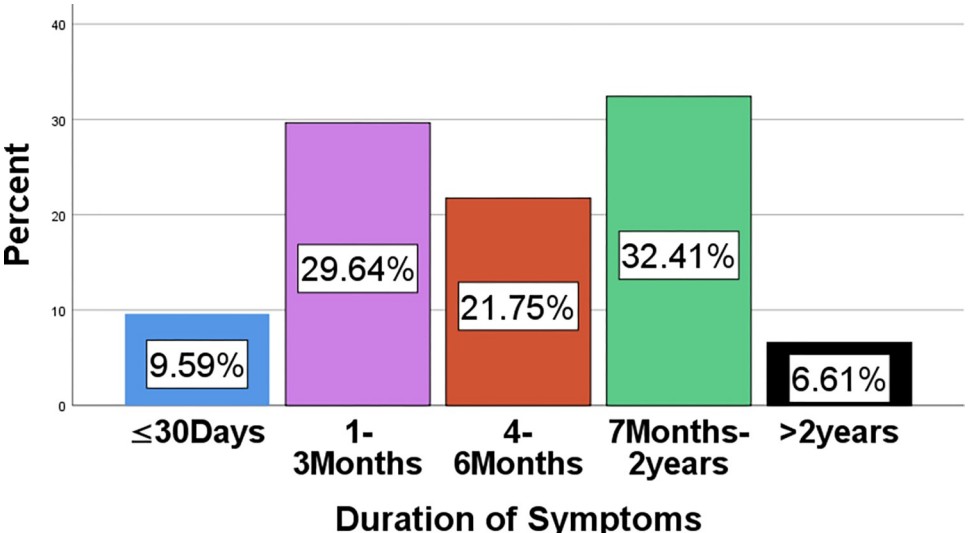

**Fig 5. Bar graph showing the distribution of the duration of symptoms in patients with cervical lesions.**

**Table 2. Type of biopsy of patients with cervical lesions.**

| Type of Biopsy/Nature of specimen | Frequency | Percent |
|---|---|---|
| Cone Biopsy | 3 | 0.6 |
| Hysterectomy | 30 | 6.4 |
| LEEP | 8 | 1.7 |
| Punch Biopsy | 428 | 91.3 |
| Total | 469 | 100.0 |

**3.1.4. Factors associated with cervical lesions.** Age ($p$ = **0.001, COR = 1.048**) and place of residence ($p$ = **0.032, COR = 0.531**) were tested at a P value less than 0.05 in bivariate logistic regression and selected as candidate variables for multivariate logistic regression. Cervical lesions were grouped into benign and HPV-related precancerous and cancerous lesions. Addresses were categorized into Jimma town, which is urban, and surrounding rural areas. Multivariate logistic regression analysis was performed on these variables using the step-wise method and the overall model was significant at a P value of 0.001. Age ($p$ = **0.001, AOR = 1.052**) and place of residency ($p$ = **0.015, AOR = 0.47**) were shown to be the independent predictors of precancerous and cancerous cervical lesions, hence older women are 1.052 times more likely to develop precancerous and cancerous cervical lesions than younger women and rural dwellers are 0.47 times more likely to develop precancerous and cancerous cervical lesions than urban dwellers which were statistically significant as shown in Tables 6–8.

## 3.2 Discussion

The most common type of biopsy performed for patients with cervical lesions was a punch biopsy, accounting for 91.3% followed by total abdominal hysterectomy accounting for 6.4%. In a study performed in India, of the total 110 cases, fifty-one were hysterectomy specimens including 25 Wertheim's hysterectomies, 16 total abdominal hysterectomies, and 10 vaginal hysterectomies. Fifty-nine specimens were cervical biopsies [20]. Additionally, in a study done in Italy, charts of 224 "low-risk" patients were retrieved and, 50 patients undergoing radical hysterectomy were matched with 100 patients undergoing open radical hysterectomy [21].

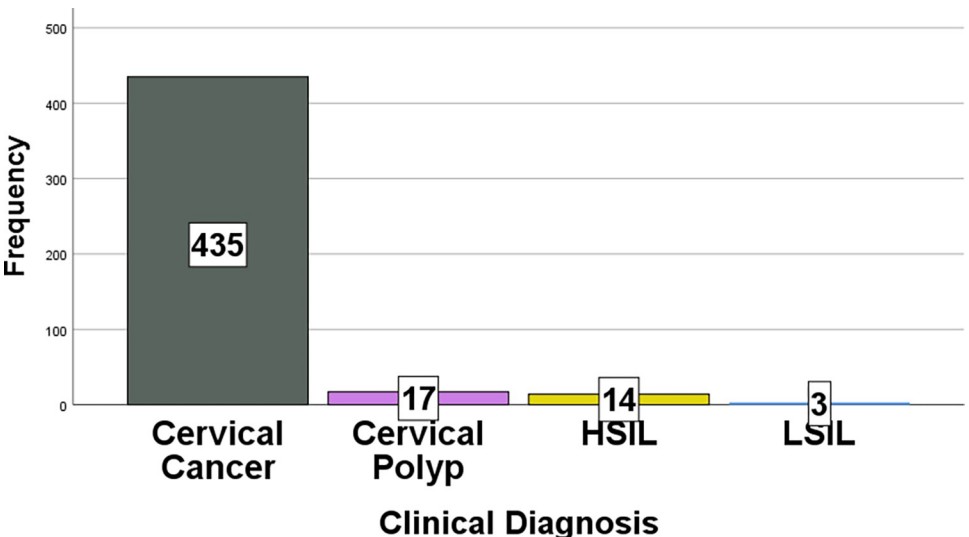

**Fig 6. Bar graph showing the distribution of clinical diagnosis of patients with cervical lesions.**

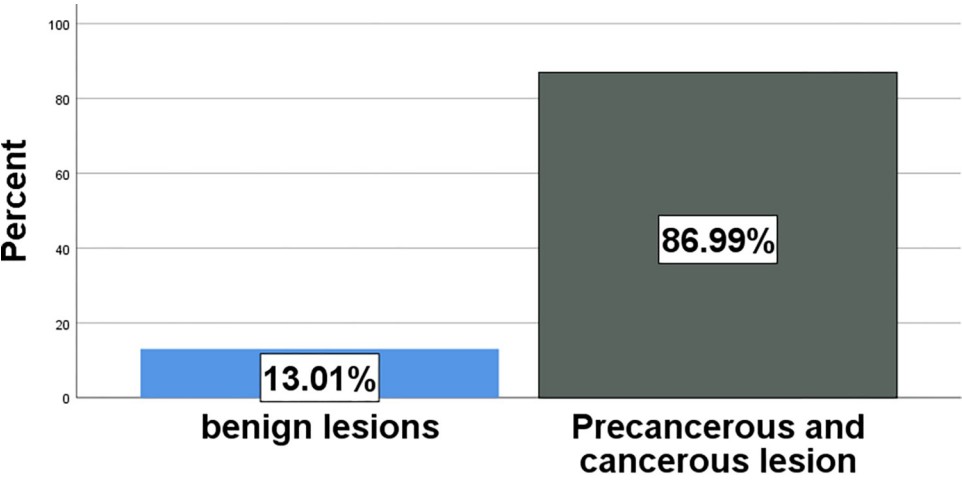

**Fig 7. Bar graph showing the distribution of cervical lesion status.**

Similarly, in a study done in Italy on precancerous cervical lesions, 2966 patients underwent conization for CIN2-3 during the study period [22]. These and other studies performed in the best centers show a higher number of hysterectomies and conizations because of the well-established chemotherapy set-up for advanced cancers and early screening and the detection of precursor lesions. In the study area, there is a higher number of punch biopsies than hysterectomies, because patients will be referred to Addis Ababa once the diagnosis of cervical cancer is made by punch biopsies.

Cervical lesions tend to affect females across a wide age range. The age distributions have a minimum value of 22 years and a maximum value of 85 years, with a mean age of 47.06 years and a standard deviation of 11.019 years. Age and place of residency were shown to be the independent predictors of precancerous and cancerous cervical lesions. This is consistent with Ameya G.'s study on 513 biopsies in Hawassa, Ethiopia, in which the age ranges were 17 to 85 years, with a mean and standard deviation of 42 and 11 years respectively [23]. Our study is also similar to a study done on 306 cervical cancer cases in Nigeria, which showed patient ages diagnosed with cervical cancer to range from 18 to 85 years, with peak occurrence in the fifth decade [24]. A study done on 1047 specimens in Lagos, Nigeria, also showed the age range of patients as 19–87 years, with a mean age of 49±13 [25]. Another study done at TASH, A.A., Ethiopia on women with histologically verified cancer of the cervix uteri also showed that the mean age was 49 years (21–91 years) [26]. Likewise, a study in Jimma, Ethiopia found that

**Table 3. Numbers and percentages of cervical lesions.**

| Type of Cervical Lesion | Frequency | Causes | Percentage |
|---|---|---|---|
| Precancerous cervical lesion | 79 | 54(68.4%) HSIL | 16.8% |
| | | 25(31.6%) LSIL | |
| Cancerous cervical lesion | 331 | See Table 4 | 70.6% |
| Noncancerous cervical lesion | 59 | 35(59.3%) Endocervical polyp | 12.6% |
| | | 18(30.5%) Squamous metaplasia | |
| | | 6(10.2%) Cervicitis | |
| Total | 469 | | 100% |

**Table 4. Histopathological classification of cervical cancer.**

| Histopathologic type | Frequency | Percentage |
| --- | --- | --- |
| Squamous cell carcinoma | 319 | 96.4% |
| Adenocarcinoma | 11 | 3.3% |
| Adenosquamous carcinoma | 1 | 0.3% |
| Total | 331 | 100% |

women aged 51–60 had increased odds of developing cervical squamous intraepithelial lesions, compared to younger women. Most women diagnosed with LSIL (40%) and HSIL (59%) were between 41–50 years old. Age was found to be a significant independent predictor of both LSIL and HSIL [27]. Our findings are consistent with this and other similar studies.

Cervical lesions range from benign inflammatory conditions and polyps to malignant conditions. Of the 469 biopsies submitted to JMC, the department of pathology, most (87%) of the specimens were precancerous and cancerous cervical lesions, while 13% were benign cervical lesions. The majority (70.6%) of cases were cancerous cervical lesions, followed by precancerous cervical lesions (16.8%) and noncancerous cervical lesions (12.6%). A study performed in Malawi on 212 cervical lesions showed that 17% had precancerous lesions, 65% were cancerous, and 18% had both precancerous and cervical cancer, making a total of 83% cancerous lesions [28]. Another study performed in India on 110 cases showed that a diagnosis of intraepithelial neoplasia was made in 32.7% of cases, and invasive carcinoma was found in 67.3% of cases [20]. Therefore, the finding of our study has a slightly higher number of cancerous cases than this last finding and other similar studies. The limited resources and services within the study area and in Ethiopia as a whole may be the reason for the differences in the accessibility of better women's healthcare practices. These include inadequate vaccination and cervical screening coverage, as well as inadequate facilities for the management of precancerous lesions.

Squamous cell carcinoma (SCC) was the most frequent cervical cancer diagnosed during the study period, accounting for 96.4% of the 333 cancerous cases, followed by adenocarcinoma (3.3%) and adenosquamous carcinoma (0.3%). This is consistent with a study done at

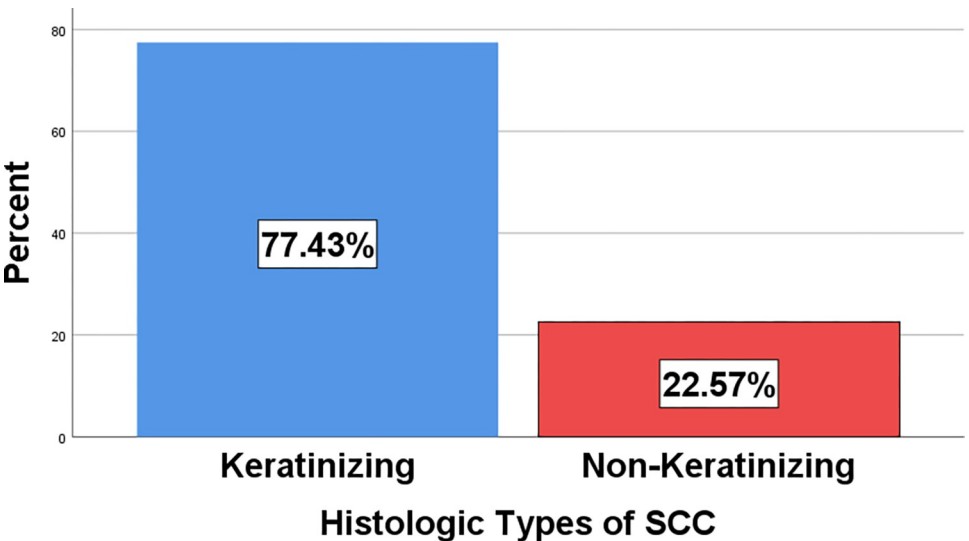

**Fig 8. Bar graph showing histologic types of SCC.**

**Table 5. Pattern of the cervical lesion by age group.**

| Age Group | Histopathologic Diagnosis | | | | | | | | Total |
|---|---|---|---|---|---|---|---|---|---|
| | Cervicitis | Squamous Metaplasia | Endocervical Polyp | LSIL | HSIL | Squamous Cell Carcinoma | Adenocarcinoma | Adenosquamous Carcinoma | |
| 20-30years | 0 | 4 | 8 | 3 | 1 | 14 | 0 | 0 | 30 |
| 31-40years | 2 | 6 | 13 | 11 | 23 | 86 | 1 | 1 | 143 |
| 41-50years | 3 | 3 | 10 | 10 | 19 | 104 | 5 | 0 | 154 |
| 51-60years | 0 | 3 | 4 | 1 | 6 | 85 | 3 | 0 | 102 |
| >60years | 1 | 2 | 0 | 0 | 5 | 30 | 2 | 0 | 40 |
| Total | 6 | 18 | 35 | 25 | 54 | 319 | 11 | 1 | 469 |

JUSH, Jimma, Ethiopia, on 154 cervical cancer cases, which showed that 91% of the cases were SCC, 5.84% were small cell carcinomas, 2.59% were adenocarcinomas, 0.64% were adenosquamous carcinomas [29]. Our study is also consistent with a study performed in Cameron on 2078 cervical cancer cases, which showed that the most common cervical cancer was SCC (81.18%), followed by adenocarcinoma (12.95%) [30]. In another study done at TASH, Addis Ababa, Ethiopia, cancer of the uterine cervix was found to be the most common malignancy among all gynecological malignancies [31]. Another study done at A.A., Ethiopia on women with postmenopausal bleeding also showed that about 94% of the cervical malignancies were squamous cell carcinoma [32]. A study done at Saint Paul Hospital Millennium Medical College (SPHMMC), Addis Ababa, Ethiopia, on breast and gynecologic malignancies also showed that SCC of the cervix was the most common type of cervical malignancy detected, accounting for 90% of all cases, while adenocarcinoma (3.85%) was the second most observed type of cervical cancer [33]. Similarly, according to a study done in Bangladesh on gynecologic malignancy, SCC was the most common histopathologic type in cervical and vulvar cancers [34].

**Table 6. Bivariate logistic regression of cervical lesion status and age.**

| Independent Variable | B-Coefficient | t-value | P-value | R square | COR |
|---|---|---|---|---|---|
| Age | 0.154 | 3.369 | 0.001 | 0.024 | 1.048 |

COR: -Crude Odds Ratio

**Table 7. Bivariate logistic regression of cervical lesion status and address.**

| Independent Variable | Beta Coefficient | t-value | P-value | R square | COR |
|---|---|---|---|---|---|
| Address | 0.099 | 2.145 | 0.032 | 0.010 | 0.531 |

COR: -Crude Odds Ratio

**Table 8. Multivariate logistic regression of cervical lesion status and associated factors.**

| Independent Variables | Beta Coefficient | t-value | P-value | AOR | R square |
|---|---|---|---|---|---|
| Age | 0.162 | 3.561 | 0.001 | 1.052 | 0.036 |
| Address | 0.111 | 2.435 | 0.015 | 0.47 | |

AOR:—Adjusted Odds Ratio

Another study done in Sudan on 372 patients showed that SCC (NOS) was the most common subtype (76.6%) and adenocarcinoma was the second most common subtype (12.6%) [35]. On the contrary, according to a study performed in India on 200 cases, 99 (49.5%) cases were malignant, and 101 (50.5%) cases were benign [36]. These last and other recent studies conducted in Western countries have shown a higher prevalence of benign cervical lesions due to early screening and detection of precursor lesions. Even though cervical cancer can be prevented through vaccination and early screening, these measures are not fully implemented in the study area. During the study period, only two women came for screening out of the 469 cases, because the PAP smear was not commenced in the study area at the time of data collection.

Of the 79 precancerous cervical conditions diagnosed during the study period, HSIL was the most frequently diagnosed precancerous lesion, accounting for 68.4%, with the other 31.6% being LSIL. This is consistent with a study performed in Hawassa, Ethiopia, on 513 biopsies, which showed that among precancerous cervical lesions, HSIL was the predominant lesion (39.8%) [23]. Similarly, a study performed in India showed that cervical intraepithelial neoplasia (CIN) 1 was diagnosed in 36.1% of patients, followed by CIN 2 (33.3%) and CIN 3 (30.6%) [20]. Likewise, a study done in northwestern Ethiopia on 335 cases, showed that cervical intraepithelial neoplasia (CIN)-1, CIN-2, and CIN-3 accounted for 40(11.9%), 12(3.6%), and 12(3.6%), respectively [37]. Our findings are consistent with this and other similar studies.

Of the 59 benign cases, endocervical polyps were the most commonly diagnosed benign lesion, accounting for 59.3%, followed by squamous metaplasia (30.5%) and cervicitis (10.2%). A study performed in Nigeria on 176 cervical specimens showed that inflammatory lesions accounted for 55 cases (59.8%) of non-neoplastic cervical lesions, followed by endocervical polyps, which accounted for 15 cases (16.3%) of non-neoplastic cervical lesions. Among the inflammatory lesions, chronic nonspecific cervicitis was the most commonly encountered lesion, constituting 40 cases (72.2%) of all inflammation [38]. In a study performed at TASH, Addis Ababa, Ethiopia chronic cervicitis was the commonest benign lesion (16.8%) [32]. Contrary to this and other similar studies, our study has a lower number of cervicitis cases, possibly because the majority of cervicitis cases are managed clinically without histopathological confirmation once a clinical diagnosis is made as per the National STD Guidelines.

## 4. Conclusion and recommendation

### 4.1. Conclusion

Cervical diseases include various disorders and occur across different age groups. The age distributions have a minimum value of 22 years and a maximum value of 85 years, with a mean age of 47.06 years. The most common presenting symptom in patients with cervical lesions was vaginal bleeding. The most common type of biopsy performed for patients with cervical lesions is a punch biopsy. Most (90%) patients with cervical complaints were clinically diagnosed with cervical cancer. Cervical lesions range from benign inflammatory conditions and polyps to malignant conditions. Most cervical lesions were precancerous and cancerous. Squamous cell carcinoma (SCC) was the most frequent type of cervical cancer. HSIL was the most frequently diagnosed precancerous cervical lesion. Endocervical polyps were the most commonly diagnosed benign cervical lesions. Precancerous and cancerous cervical lesions tend to increase with increasing age and in rural dwellers.

### 4.2. Recommendation

There was a rising burden of cervical cancer. This implies that there is a need to educate the community to improve health-seeking behavior and on possible preventive strategies for

cervical cancer. It is also strongly recommended that JMC's Department of Pathology commence cervical cytological examinations to detect dysplasia before it turns into full-blown cancer. We recommend that the Federal Ministry of Health strengthen the ongoing nationwide HPV vaccinations, screenings, and interventions for cervical cancer at a much faster rate.

## Supporting information

**S1 Checklist.**
(PDF)

**S1 Checklist. STROBE statement—checklist of items that should be included in reports of observational studies.**
(PDF)

**S1 File.**
(SAV)

**S2 File.**
(DOCX)

## Acknowledgments

I thank Jimma University for allowing me to undertake the research. I am grateful to Jimma Medical Center, Department of Pathology, for providing me with the necessary resources. I also thank my advisors and friends for giving me constructive comments and important information to prepare my proposal and thesis. Last, my warm gratitude goes to all my family members for providing me with emotional support during this study.

## Author Contributions

**Conceptualization:** Birhanu Hailu Tirkaso.

**Data curation:** Tewodros Wubshet Desta.

**Formal analysis:** Birhanu Hailu Tirkaso, Tesfaye Hurgesa Bayisa.

**Investigation:** Birhanu Hailu Tirkaso.

**Methodology:** Birhanu Hailu Tirkaso.

**Project administration:** Birhanu Hailu Tirkaso.

**Resources:** Birhanu Hailu Tirkaso.

**Software:** Birhanu Hailu Tirkaso.

**Validation:** Tewodros Wubshet Desta.

**Writing – original draft:** Birhanu Hailu Tirkaso.

**Writing – review & editing:** Tesfaye Hurgesa Bayisa.

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
