## [Decision Letter · Decision Letter 0]

4 Jul 2023

PONE-D-23-17869Histopathologic patterns and factors associated with cervical lesions at Jimma Medical Center, Jimma, Southwest Ethiopia: A two-year crosssectional studyPLOS ONE

Dear Dr. Tirkaso,

Thank you for submitting your manuscript to PLOS ONE. After careful consideration, we feel that it has merit but does not fully meet PLOS ONE’s publication criteria as it currently stands. Therefore, we invite you to submit a revised version of the manuscript that addresses the points raised during the review process.

We look forward to receiving your revised manuscript.

Kind regards,

Andrea Giannini

Academic Editor

PLOS ONE

Journal Requirements:

- https://www.iccp-portal.org/sites/default/files/plans/NCCP%20Ethiopia%20Final%20261015.pdf

- http://ispub.com/IJPA/10/2/4126#

- http://10.140.5.162//handle/123456789/4080

In your revision ensure you cite all your sources (including your own works), and quote or rephrase any duplicated text outside the methods section. Further consideration is dependent on these concerns being addressed.

3. We noted in your submission details that a portion of your manuscript may have been presented or published elsewhere. [Yes,this paper is based on the thesis of Birhanu Hailu, the corresponding author. It has been on the institutional website of Jimma University:  https://repository.ju.edu.et/handle/123456789/4080 and this is also mentioned on the disclosure section.] Please clarify whether this [conference proceeding or publication] was peer-reviewed and formally published. If this work was previously peer-reviewed and published, in the cover letter please provide the reason that this work does not constitute dual publication and should be included in the current manuscript.

Additional Editor Comments:

Dear authors,

the topic of the present article titled “Histopathologic patterns and factors associated with cervical lesions at Jimma Medical Center, Jimma, Southwest Ethiopia: A two-year crosssectional study” is very interesting, the paper and the aim falls within the scope of the journal but the article needs major improvements.

The introduction, material and method section and tables should be modified and improved.

The manuscript should be organized better and English should be improved.

I suggest improving the manuscript with the reviewers' comments.

Reviewers' comments:

Reviewer's Responses to Questions

**Comments to the Author**

1. Is the manuscript technically sound, and do the data support the conclusions?

Reviewer #1: Yes

Reviewer #2: No

2. Has the statistical analysis been performed appropriately and rigorously? 

Reviewer #1: Yes

Reviewer #2: Yes

3. Have the authors made all data underlying the findings in their manuscript fully available?

Reviewer #1: Yes

Reviewer #2: Yes

4. Is the manuscript presented in an intelligible fashion and written in standard English?

Reviewer #1: Yes

Reviewer #2: Yes

5. Review Comments to the Author

Reviewer #1: The article brings an important topic with relevant data. Tables and graphs are adequate and very interesting. Personally, I don't really like pie charts, it's just a recommendation, as I don't see this profile being used much in articles.

Reviewer #2: In my opinion, the analyzed topic is interesting enough to attract the readers’ attention. I think that the abstract of this article is very clear and well structured.

In my opinion, the discussion could be studied in depth and extended. Maybe, it could be useful the evaluation of the long term outcomes of this condition in order to evaluate all the aspects of cervical cancer. In particular, I suggest these articles to get deeper in the topic: PMID: 36992282 and PMID: 37149905. Because of these reasons, the article should be revised and completed. Considered all these points, I think it could be of interest for the readers and, in my opinion, it deserves the priority to be published after minor revisions.

6. PLOS authors have the option to publish the peer review history of their article (what does this mean?). If published, this will include your full peer review and any attached files.

Reviewer #1: **Yes: **Renata Mirian Nunes Eleutério

Reviewer #2: No

---

## [Author Response · Author response to Decision Letter 0]

11 Aug 2023

Response to Academic Editor:-Thank you very much for your insightful comments and I have tried to address the comments as requested.

1.The manuscript is now revised according to PLOS ONE's style requirements in accordance with the formatting guidelines provided. We have updated the title page and renamed the chapters as outlined in the PLOS ONE guidelines. Figures were also formatted according to the guidelines and included in the main document. The figures will also be uploaded as a separate file in case they are needed during manuscript processing.

2.Regarding the minor occurrence of overlapping text with previous publications, we revised the specific paragraphs and rephrased the text to avoid plagiarism and we have cited the paper mentioned by the academic editor. Please see pages 1-3. The revised manuscript is checked for plagiarism using online checkers and checked for similarity index using Turnitin before resubmission. The majority of the similarity index came from the same thesis of the corresponding author put on the institutional website by Jimma University to avoid topic redundancy for future graduates but was not formally published. One thing, we want to clarify is that this manuscript is the refined version of the manuscript posted on Jimma University's website: https://repository.ju.edu.et/handle/123456789/4080.

3.Concerning dual publications, the manuscript was not peer-reviewed and formally published anywhere. As a part of second-degree completion, Jimma University put the thesis of postgraduate graduates on its website so that each graduate student selects a unique research topic.We put the link to the institutional repository to indicate that our work is based on the material posted on the website. Because of this, this work does not constitute a dual publication.

4.The other issue raised was the data availability statement. All relevant data are within the manuscript and its Supporting Information files. Please update this for us.

Response to Reviewer #1:-Thank you so much for your nice comments,Pie charts have been replaced by bar graphs.

Response to Reviewer #2: Thank you for your in-depth revision.The discussion part has been expanded now to include the type of biopsies (Nature of specimen) by incorporating additional references. Because the focus of this thesis is to show the magnitude of the cervical cancer from Pathology point of view,it will be difficult for us to go beyond this to the details of surgical management options due to the limited set up.I am eternally grateful.

---

## [Decision Letter · Decision Letter 1]

9 Jan 2024

PONE-D-23-17869R1Histopathologic patterns and factors associated with cervical lesions at Jimma Medical Center, Jimma, Southwest Ethiopia: A two-year crosssectional studyPLOS ONE

Dear Dr. Tirkaso,

Thank you for submitting your manuscript to PLOS ONE. After careful consideration, we feel that it has merit but does not fully meet PLOS ONE’s publication criteria as it currently stands. Therefore, we invite you to submit a revised version of the manuscript that addresses the points raised during the review process.

We look forward to receiving your revised manuscript.

Kind regards,

Andrea Giannini

Academic Editor

PLOS ONE

Journal Requirements:

Additional Editor Comments:

In order to ensure that the methodological aspects of the study are fully evaluated, an additional reviewer was sought for this round of review. A higher than usual number of reviewers accepted this invitation. Nevertheless, some critical feedback was obtained, and I would be grateful if you could please particularly address the specific concerns raised by Reviewers 5 and 6.

Reviewers' comments:

Reviewer's Responses to Questions

**Comments to the Author**

1. If the authors have adequately addressed your comments raised in a previous round of review and you feel that this manuscript is now acceptable for publication, you may indicate that here to bypass the “Comments to the Author” section, enter your conflict of interest statement in the “Confidential to Editor” section, and submit your "Accept" recommendation.

Reviewer #2: All comments have been addressed

Reviewer #3: All comments have been addressed

Reviewer #4: All comments have been addressed

Reviewer #5: (No Response)

Reviewer #6: All comments have been addressed

Reviewer #7: All comments have been addressed

Reviewer #8: All comments have been addressed

2. Is the manuscript technically sound, and do the data support the conclusions?

Reviewer #2: Yes

Reviewer #3: Yes

Reviewer #4: Partly

Reviewer #5: Partly

Reviewer #6: Yes

Reviewer #7: Yes

Reviewer #8: Partly

3. Has the statistical analysis been performed appropriately and rigorously? 

Reviewer #2: Yes

Reviewer #3: Yes

Reviewer #4: Yes

Reviewer #5: No

Reviewer #6: Yes

Reviewer #7: Yes

Reviewer #8: Yes

4. Have the authors made all data underlying the findings in their manuscript fully available?

Reviewer #2: Yes

Reviewer #3: Yes

Reviewer #4: Yes

Reviewer #5: Yes

Reviewer #6: Yes

Reviewer #7: Yes

Reviewer #8: No

5. Is the manuscript presented in an intelligible fashion and written in standard English?

Reviewer #2: Yes

Reviewer #3: Yes

Reviewer #4: No

Reviewer #5: Yes

Reviewer #6: Yes

Reviewer #7: Yes

Reviewer #8: Yes

6. Review Comments to the Author

Reviewer #2: The quality of the manuscript has improved thanks to the changes made. I think it could be of interest to the readers and, in my opinion, it deserves the priority to be published.

Reviewer #3: Thank you for going through the manuscript and the reviewers' points

The manuscript is well written and falls within the aim of this Journal.

Reviewer #4: I read this manuscript, I found it interesting. However, to accept this publication, authors have to:

Major points

- Use other techniques to confirm their results or

- Increase the number of patients use in this studies

Minor Points

Authors have to improve grammar and typos in their manuscript

Overall, the manuscript is interesting and could be consider as a new window to other researchers to study the molecular mechanisms involved.

Reviewer #5: The authors have raised a very important public health problem in the study setting. Howevere,there are some flaws in the methodology employed.

1. would be nice if representative sample is used, which will enhance the rigor of the manuscript

2. The data collection period of May 1 to June 30, 2019, and the study period of September 12, 2018, to September 12, 2020, is confusing.

3. You need to calculate sample size for the third specific objective.

4. your study is based on the pathology report data, how did you collect the clinical data

5. Stastical analysis need major revision in scientifically and stastically readable and understandable fashion.

6. Authors mentioned some limitations of the study, what was the strength of this study.

7. Only 2 of the 469 patients came for screening purpose, this is a very important figure which should be elaborated further.

Reviewer #6: am appreciate the objective of this article. however 1.the methodological part have week explanation 2. you made selected sample from population but among several sampling method which you used can not be expressed in this article. 3. there no any formula show how to select sample 4. in the methodology part i.e for the logistic model have weak explanation or formula to be used. 5. in result part you made give a conclusion based on descriptive result was not correct.

Reviewer #7: The authors have revised manuscript as per suggestions and corrections; therefore, It can be published in its present form.

Reviewer #8: EVERTHING IS Good with slight modification. but The problem is not well stated. Please show the real gap why study analysis is important in this study? - Why Model diagnostics is important for your study?

- Then describe different types of model diagnostic methods and why one is used over other or what is the importance of each of them. Then give your source (Reference).

- No need for mathematical methods to describe.

- Try to contextualize each of the model parameters and coefficients.

7. PLOS authors have the option to publish the peer review history of their article (what does this mean?). If published, this will include your full peer review and any attached files.

Reviewer #2: No

Reviewer #3: No

Reviewer #4: **Yes: **Armel Herve Nwabo Kamdje

Reviewer #5: No

Reviewer #6: No

Reviewer #7: No

Reviewer #8: No

---

## [Author Response · Author response to Decision Letter 1]

7 Feb 2024

1. We would like to thank the editor for inviting additional reviewers for an in-depth review of the manuscript. We updated our reference list papers by removing papers that have been retracted (Atul J. RJ, IE al. Histopathological study of tumors of Cervix at MGM Medical College. Medical College of India. 2014;1(1) and Nigatu B, Gebrehiwot Y, Kiros K, Ergete W. Number 1 Ethiopian Journal of Reproductive Health) and replacing them with relevant current references which are highlighted on the revised manuscript, to ensure that it is complete and correct.

2. Thank you very much Reviewer 2 for acknowledging the improvement and recommending the manuscript for publication, I am eternally grateful.

3. Thank you, Reviewer 3, for your evaluation and confirming that our manuscript falls within the aim of the Journal and is well-written. I am extremely grateful. 

4. Thank you, Reviewer 4, for your review. I am grateful and have addressed your comments below.

The findings of our study are quite similar to those from developing countries. This is mainly because the patients in our study tended to present late and were not vaccinated against the Human Papillomavirus (HPV). To establish detailed causal relationships between the findings and other factors, further studies involving genetic and molecular testing would be required. However, such studies are beyond the scope of our present research.

Although it is generally accepted that a larger sample size of patients would lead to more reliable results, in this particular case, we only had access to well-documented hard-copy biopsy reports that were two years old at the time of data collection. Therefore, all 469 eligible reports were included in the study.

The document underwent a thorough review for grammar and punctuation by both language software and a colleague who is proficient in written English. 

5. Thank you, Reviewer 5, for your detailed review of the manuscript. I am grateful and have outlined my responses below.

The idea of using representative sampling is valid and initially, we planned to use simple random sampling to select a representative sample of biopsy reports from the available 543 reports. The expected sample size was 226, after adjusting using a correction formula, but we later decided to include all 469 eligible reports (see Figure 1, page 6) in the study based on the advice of our colleagues from the Epidemiology side and a review of the literature, as outlined in our manuscript's references. This approach gave us a margin error of 1.67%, which we deemed acceptable. We were aware that having a sample that is too big or too small can waste resources or lead to unreliable results. We also needed to include small but significant factual information like the two cases of cervical screening, which made the sample size of 469 feasible for resource management. However, reviewer number 4 suggested increasing the study units, which we agree with, but there were no well-documented biopsy reports other than 2017/18 and 2018/19 at the time of the study.

The previous dates mentioned in our manuscript have been corrected. The period of 12th September 2018 to 11th September 2020 has been adjusted to 12th September 2017 to 12th September 2019, and the data collection period has been corrected from May 1 to June 30, 2019, to May 1 to June 30, 2020. I apologize for any confusion this may have caused. The mistake occurred when I converted the Ethiopian calendar into the Gregorian calendar. I also want to clarify that the data was collected from two-year reports. It is important to note that the Ethiopian calendar lags behind the Gregorian calendar by 7 to 8 years, depending on the month, and that all activities of the pathology department are renewed in September, which marks the beginning of a new year in Ethiopia. Thank you for bringing this to our attention. 

I have noted your request regarding the issue of calculating the sample size for the third specific objective. Please note that our study is descriptive and we are not establishing causality. We are simply observing that certain factors that have been well-studied in the literature and textbooks could influence the frequency of cervical cancer, which varies with age group. This pattern has been reported in several studies, including one conducted by Ameya and his colleague from Hawassa, Ethiopia, which matches our report (Ameya & Yerakly, 2017). Another study conducted in Jimma, Ethiopia, also revealed that women above 50 years old were more likely to have advanced cervical lesions (Tesfaw et al., 2020). We have cited these papers as examples from the references list in our manuscript (see reference numbers 22 and 26) because they have been published in well-known journals. We encourage the comparison of the methodology parts of the second article, as it includes bivariate and multivariate logistic regressions, and was done on the same study area of our paper. 

Our research is based solely on the pathology report data. We were unable to collect clinical data from patient cards as we do not have access to their medical records without consent from the medical record service areas. However, the biopsy request format of Jimma Medical Center, which is included as supporting documents, contains all the variables necessary for our study. I created the questionnaire based on the biopsy requests and reports. It is important to note that we did not include all aspects of cervical cancer, such as parity, marital status, sexual history, HIV status, etc. This is why the title of our study is Histopathologic Patterns, rather than Clinicopathologic Patterns. See Biopsy Form of JMC.

I understand that there is a comment regarding the statistical analysis of our paper that needs major revision. However, we have already undergone a rigorous review process and have presented refined versions of the statistical output. Additionally, the methodology section of our report is now expanded to include the logistic regression model and the formula we used. To run the regression, we utilized SPSS software. However, we did not include detailed statistical and analytical steps in the manuscript, as we presumed that readers understand how the output results are obtained. Also, we received feedback from reviewer 8 who suggested that we contextualize the model instead of providing a technical explanation. Our findings indicate that age and place of residency are independent predictors of precancerous and cancerous cervical lesions. This is clearly stated on page 16 of our paper, and all SPSS values have been included in the tables. In addition, we have added more references to the discussion section. If you could provide us with specific guidance on statistical issues such as P-values, COR, and AOR, we would be happy to make changes or provide explanations for our interpretations. Thank you. See pages 6,7,16, 17, and 19

We have acknowledged some limitations of our study to ensure that readers consider them when interpreting the results. However, we have emphasized the strength of the study by providing a detailed description of the study area, including its large catchment area and the availability of pathology services. During the planning phase, we conducted a SWOT analysis and developed a conceptual framework, but we have only included selected concepts in this article based on the journal's requirements. 

I appreciate your recognition of the fact that only two out of the 469 patients came for screening purposes in our study. This was because PAP smear tests weren't available in the study area at the time of data collection. However, our study findings have been submitted to JMC's Department of Pathology with a strong recommendation to commence cervical cytological examinations to detect dysplasia before it turns into full-blown cancer. As a result, the department has started providing PAP services now, and we expect the situation to be different in future research in the study area. See page 19

6. Thank you, Reviewer 6, for appreciating the objective of the study and for your insightful comments, the responses and explanations are provided below.

The idea of using sampling and sampling formula is valid and initially, we planned to use Yamane’s formula and simple random sampling to select a representative sample of biopsy reports from the available 543 reports. The expected sample size was 226, after adjusting using a correction formula, but we later decided to include all 469 eligible reports (see Figure 1, page 6) in the study based on the advice of our colleagues from the Epidemiology side that most of the published papers on medical literature tend to include all eligible cases as the study unit if it is feasible as outlined in our manuscript's references. We have cited these papers as examples from the references list in our manuscript (see reference numbers 22 and 26) because they have been published in well-known journals. This approach gave us a margin error of 1.67%, which we deemed acceptable. We were aware that having a sample that is too big or too small can waste resources or lead to unreliable results. We also needed to include small but significant factual information like the two cases of cervical screening, which made the sample size of 469 feasible for resource management. However, reviewer number 4 suggested increasing the study units, which we agree with, but there were no well-documented biopsy reports other than 2017/18 and 2018/19 at the time of the study.

The methodology section of our report is now expanded to include the logistic regression model and the formula we used. To run the regression, we utilized SPSS software. However, we did not include detailed statistical and analytical steps in the manuscript, as we presumed that readers understand how the output results are obtained. Additionally, we received feedback from reviewer 8 who suggested that we contextualize the model instead of providing a technical explanation. See page 7

In the conclusion part, the phrase ‘‘those coming from peripheral areas’’ is replaced with ‘‘rural dwellers’’ for clarity. See page 20

7. Thank you very much Reviewer 7 for acknowledging the improvement and recommending the manuscript for publication, I am eternally grateful.

8. Thank you, Reviewer 8, for evaluating the manuscript and for your insightful comments, and the responses and explanations are provided below.

The problem is stated in the rationale of the study and the significance of the study subheadings of the introduction. See pages 3 and 4

The methodology section of our report is now expanded to include the regression model we used and the reference for it. See page 7

---

## [Decision Letter · Decision Letter 2]

19 Mar 2024

Histopathologic patterns and factors associated with cervical lesions at Jimma Medical Center, Jimma, Southwest Ethiopia: A two-year cross-sectional study

PONE-D-23-17869R2

Dear Dr. Tirkaso,

We’re pleased to inform you that your manuscript has been judged scientifically suitable for publication and will be formally accepted for publication once it meets all outstanding technical requirements.

Kind regards,

Andrea Giannini

Academic Editor

PLOS ONE

Additional Editor Comments (optional):

The manuscript has been modified with the comments of the reviewers. It is now ready to be published.

Reviewers' comments:

Reviewer's Responses to Questions

**Comments to the Author**

1. If the authors have adequately addressed your comments raised in a previous round of review and you feel that this manuscript is now acceptable for publication, you may indicate that here to bypass the “Comments to the Author” section, enter your conflict of interest statement in the “Confidential to Editor” section, and submit your "Accept" recommendation.

Reviewer #4: All comments have been addressed

Reviewer #5: (No Response)

Reviewer #7: All comments have been addressed

2. Is the manuscript technically sound, and do the data support the conclusions?

Reviewer #4: Yes

Reviewer #5: (No Response)

Reviewer #7: Yes

3. Has the statistical analysis been performed appropriately and rigorously? 

Reviewer #4: Yes

Reviewer #5: (No Response)

Reviewer #7: Yes

4. Have the authors made all data underlying the findings in their manuscript fully available?

Reviewer #4: Yes

Reviewer #5: (No Response)

Reviewer #7: Yes

5. Is the manuscript presented in an intelligible fashion and written in standard English?

Reviewer #4: Yes

Reviewer #5: (No Response)

Reviewer #7: Yes

6. Review Comments to the Author

Reviewer #4: your manuscript intitled ``Histopathologic patterns and factors associated with cervical lesions at Jimma Medical Center, Jimma, Southwest Ethiopia: A two-year cross-sectional study`` is interesting and can be accepted in this journal.

Reviewer #5: (No Response)

Reviewer #7: Revisions are incorporated properly therefore, I give the recommendation for publication in its present form.

7. PLOS authors have the option to publish the peer review history of their article (what does this mean?). If published, this will include your full peer review and any attached files.

Reviewer #4: **Yes: **Armel Herve Nwabo Kamdje

Reviewer #5: No

Reviewer #7: No

---

## [Editor Report · Acceptance letter]

24 Mar 2024

PONE-D-23-17869R2 

PLOS ONE

Dear Dr. Tirkaso, 

I'm pleased to inform you that your manuscript has been deemed suitable for publication in PLOS ONE. Congratulations! Your manuscript is now being handed over to our production team.

Kind regards, 

on behalf of

Dr. Andrea Giannini 

Academic Editor

PLOS ONE